# Construction and validation of an ultraviolet germicidal irradiation system using locally available components

Eric Schnell[1,2]*, Elham Karamooz[3,4], Melanie J. Harriff[4,5], Jane E. Yates[5], Christopher D. Pfeiffer[6,7], Stephen M. Smith[3,4]

**1** Operative Care Division, VA Portland Health Care System, Portland, OR, United States of America,
**2** Department of Anesthesiology and Perioperative Medicine, OHSU, Portland, OR, United States of America, **3** Pulmonary and Critical Care Medicine, VA Portland Health Care System, Portland, OR, United States of America, **4** Pulmonary and Critical Care Medicine, OHSU, Portland, OR, United States of America, **5** Research and Development, VA Portland Health Care System, Portland, OR, United States of America, **6** Infectious Diseases Section, VA Portland Health Care System, Portland, OR, United States of America, **7** Division of Infectious Diseases, OHSU, Portland, OR, United States of America

* schneler@ohsu.edu

**Data Availability Statement:** All relevant data are within the manuscript and its Supporting Information files.

## Abstract

Coronavirus disease (COVID-19), the disease caused by the severe acute respiratory syndrome coronavirus 2 (SARS-CoV-2) virus, is responsible for a global pandemic characterized by high transmissibility and morbidity. Healthcare workers (HCWs) are at risk of contracting COVID-19, but this risk has been mitigated through the use of personal protective equipment such as N95 Filtering Facepiece Respirators (FFRs). At times the high demand for FFRs has exceeded supply, placing HCWs at increased exposure risk. Effective FFR decontamination of many FFR models using ultraviolet-C germicidal irradiation (UVGI) has been well-described, and could maintain respiratory protection for HCWs in the face of supply line shortages. Here, we detail the construction of an ultraviolet-C germicidal irradiation (UVGI) device using previously existing components available at our institution. We provide data on UV-C dosage delivered with our version of this device, provide information on how users can validate the UV-C dose delivered in similarly constructed systems, and describe a simple, novel methodology to test its germicidal effectiveness using in-house reagents and equipment. As similar components are readily available in many hospitals and industrial facilities, we provide recommendations on the local construction of these systems, as well as guidance and strategies towards successful institutional implementation of FFR decontamination.

## Introduction

One emergent challenge during the current COVID-19 pandemic has been that the SARS-CoV-2 virus frequently infects health care workers, threatening to deplete the pool of people available to care for sick patients at a time when they are most needed. Respiratory protection

**Funding:** The authors would like to acknowledge grant support by the Department of Veterans Affairs, Veterans Health Administration, Office of Research and Development, Biomedical Laboratory Research and Development Merit Review Awards I01-BX004938(ES), I01-CX001562 (MH), I01-BX002547 (SMS), a Department of Defense CDMRP Award W81XWH-18-1-0598 (ES), NIH NINDS 1R21-NS102948 (Koerner/ES), NIH NIGMS R01-GM134110 (SMS), NIH R01-AI129976 (MH) and 1R21-AI151079-01 (EK).

**Competing interests:** The authors have declared that no competing interests exist.

is an important component of interrupting the chain of SARS-CoV-2 transmission. The Centers for Disease Control and Prevention (CDC) recommends N95 Filtering Facepiece Respirators (FFRs) or the equivalent for those engaged in the care of SARS-CoV-2 positive patients [1]. However, N95 FFRs have been in critically short supply due to the intersection of dramatically increased global demand and fractured supply chains, presenting an urgent need for the implementation of decontamination strategies for used N95 FFRs during this crisis.

N95 FFRs can be decontaminated and re-used several times without loss of protective efficacy [2–9]. Although a wide variety of modalities effectively decontaminate N95 FFRs, some of these modalities (such as spraying with alcohol) degrade the mask material's aerosol filtration abilities or alter subsequent FFR fit [4]. Other modalities, such as vaporized hydrogen peroxide, are highly effective and preserve FFR function [4], but require equipment that could be difficult to obtain and operationalize during a pandemic. Ultraviolet-C germicidal irradiation (UVGI) is a mode of N-95 FFR decontamination that is effective and more easily set-up. After the 2003 severe acute respiratory syndrome (SARS) epidemic, numerous studies demonstrated that coronaviruses similar to SARS-CoV-2, are inactivated by light in the UV-C spectrum (200-280nm) [6]. Inactivation occurs by photochemical degradation of the coronavirus genetic material which is composed of single-stranded RNA (ssRNA) [10, 11]. Institutions have developed protocols to employ commercial light sources to perform UVGI-based N95 FFR decontamination [10]. However, commercial UVGI systems are costly and in short supply during the current COVID-19 pandemic.

Here, we outline how we developed and tested an inexpensive system for UVGI-based N95 FFR decontamination and confirmed its germicidal efficacy. Furthermore, we discuss technical considerations, practical guidance, and alternative approaches to outline how to rapidly construct a system to permit decontamination of N95 FFRs and thus attenuate the impact of critical PPE shortages.

## Methods

### UVGI chamber design and construction

Light sources capable of producing UV-C light are normally widely available. During the COVID-19 pandemic, many "implementation ready", commercial UVGI systems became unavailable, so we designed and constructed an in-house UVGI system, using equipment already available on site. Apart from biological safety hoods, UV-C bulbs are used in HVAC and water decontamination applications, and are frequently available from a hospital's physical plant. A variety of bulbs exist for this purpose, which can vary both in power as well as in terms of far UV-C wavelength production (175–210 nm). Far UV-C light can generate hazardous ozone gas [11], and additional design considerations would be necessary if these wavelengths were produced.

We designed and built a chamber specifically for UVGI decontamination of N95 FFRs using a local supply of low-pressure UV-C bulbs (Philips TUV G30T8 30W / 35 inch bulbs; dominant emission at 254 nm, minimal emission at 175–210 nm). Our design consisted of two planar arrays of UV-C bulbs facing each other to produce an adequate field of UV illumination to both sides of each mask and allow for simultaneous decontamination of multiple masks (**Fig 1**). Our electrical shop assembled the arrays using standard low pressure light mounts ('tombstones') and regulated power supplies (ballasts; see **Table 1** for materials). Each array was 4 bulbs wide (12 cm between centers of adjacent bulbs) and 3 bulbs long end-to-end (12 bulbs per array), and the distance between arrays was 70 cm. FFRs are suspended from stretches of nylon line (Black and Decker), strung between two hooks with a thick rubber band at one end to maintain tension. Although not performed at our facility, this system could be

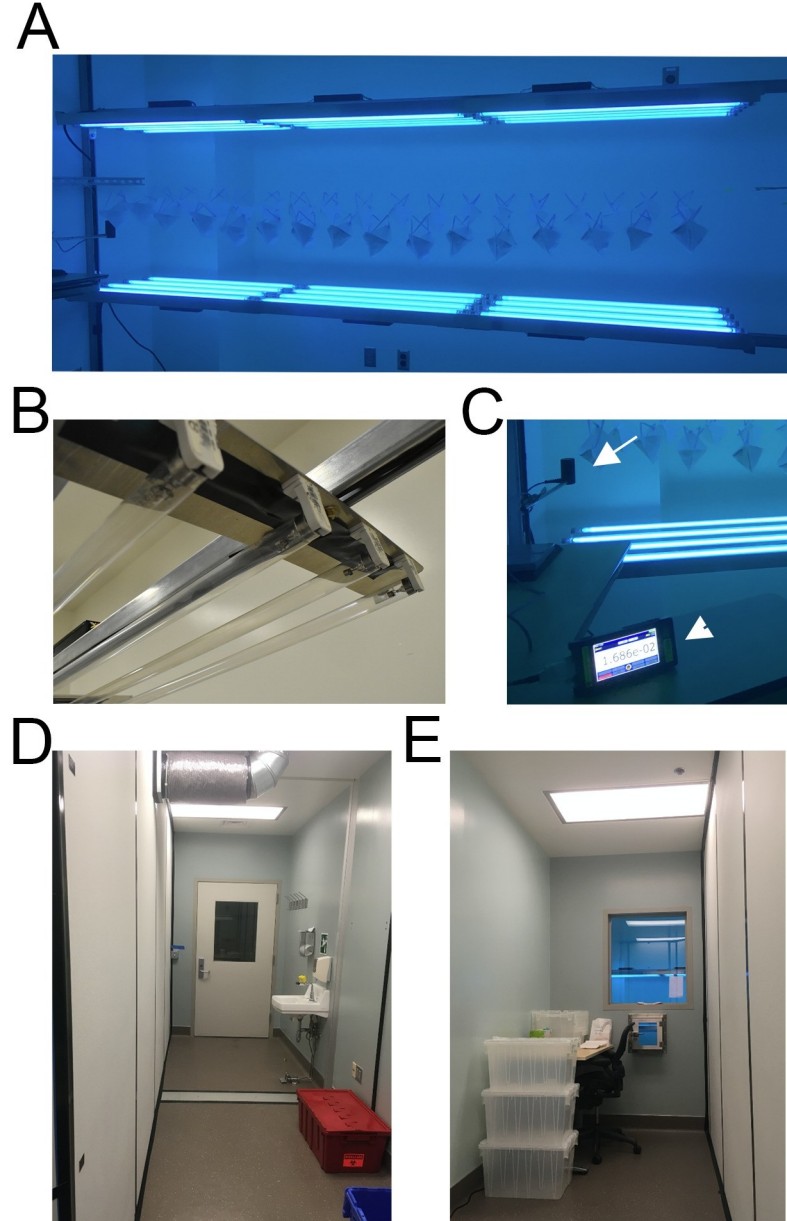

**Fig 1. UVGI arrays.** A. Each array consists of 12 UV-C bulbs in a 3x4 array, facing each other, with masks suspended between arrays. B. Longitudinal end of one array showing lampholder mounts on aluminum stock. C. Light dosimeter probe on left (white arrow) mounted to measure light at iso-irradiant site relative to lowest exposure of each mask. Light meter console (arrowhead) in foreground for monitoring total dose delivered (as seen from clean workroom). D. "Dirty" workroom entry to UVGI Suite. E. "Clean" workroom for packaging decontaminated masks.

scaled up by increasing the number of paired 4-bulb arrays, in a linear or side-by-side arrangement, with consideration to facilitate staff access for FFR loading/unloading.

Light fixtures were mounted onto flat aluminum stock bases, with mounting bars bracketed across an 8 x 10 ft room and bolted to walls (**Fig 1**). An additional safety rail was constructed on the operator side of the array to protect bulbs and staff from inadvertent contact. UV-C exposure is harmful to skin and eyes, so a UV-C-blocking barrier (glass) was used to protect staff from UV-C exposure during decontamination, while still allowing UVGI technicians to

**Table 1. Construction materials for UV-C bulb arrays.**

| **Bulb arrays:** |
| --- |
| 24x Philips TUV G30T8 30W / 35 inch germicidal bulbs (or equivalent). |
| 48x Fluorescent Lampholder, T8, Bi-Pin ("tombstone"). |
| 6x Electronic Ballast, Fluorescent, Input Voltage: 120-277V, Instant Start, Centium Series. |
| #18 copper conductor (red, yellow, blue, black, white) to wire lampholders to ballast. |
| Flat aluminum stock (for lampholder mounting) |
| SO cord #12–30' |
| 1 male cord cap |
| 1–4" junction box and cover |
| 4-Cord grips for SO cord |
| 3 port wago connectors |
| 1-20A switch and single gang box |
| **Mounting Hardware:** |
| Unistrut, 3-deep channel, 40', and 4–90 degree 4 hole brackets |
| 20'-Aluminum angle, 8-Flat aluminum stock 4"X21" |
| Black duct tape to cover wires in the Unistrut |
| 6–32 screws to mount lampholders to flat aluminum stock |
| ¼" toggle anchors to mount deep strut to walls |
| ¼" strut hardware to mount fixtures to the strut |

observe the light meter readings. UV-C light did not penetrate the glass, as verified by our UV-C dosimeter.

## UVGI intensity calibration and measurement

To measure UV-C dose, we used the ILT2400-UVGI meter (International Light Technologies, Peabody, MA), which is selectively sensitive to the wavelengths in the UV-C spectrum and reports UV-C light irradiance in $W/cm^2$. This dosimeter can also integrate light exposure in real time, and report total UV-C dose delivered in $J/cm^2$. We mounted this detector inside of our UVGI chamber to verify effective light delivery with each UVGI decontamination cycle, and positioned it to measure light intensity in a direction/location corresponding to the least amount of UV-C energy that would be received by any FFR component (see below), thus ensuring that all sections of the mask receive adequate irradiation. We strongly recommend irradiance/energy measurement for newly assembled systems and during each cycle as a quality control parameter, as UV-C bulb output can vary between bulbs, within the same bulbs as they "burn-in" and age, and even during use (in our hands, irradiance increased substantially over the first minute of being turned on). An optimal UV-C dosimeter must be calibrated, which can be verified through ISO17025 accreditation (quality assurance) and NIST traceability (calibrated to a known source), and ideally indicate both irradiance and total energy delivered for ease-of-use. However, although UV-C dosimeters are frequently available in larger-scale industrial hygiene programs, their availability might also be subject to resource availability and cost constraints. This has led to the evaluation of UV-C sensitive photochromic paper as a low-cost alternative methodology for the quantitative evaluation of UVGI processes [12].

## Locally-produced bactericidal assay

Glass coverslips (Fisherbrand 18CIR-1.5) were sterilized in 3% bleach and rinsed in 70% ethanol in a biosafety cabinet with laminar flow. DH5α *Escherichia coli* (*E. coli*) frozen at -80C in 10% glycerol was grown in sterile Luria broth (LB) until bacterial concentration following

10-fold dilution reached an optical density 600 ($OD_{600}$) of 0.25–0.4 (Amersham Biosciences Ultrospec 10 spectrophotometer). $OD_{600}$ was recorded for each run to limit variation between experiments. Biological indicators (BIs) were made by plating 4 μL of *E. coli*–containing suspension onto sterilized coverslips in a biosafety cabinet and allowing them to dry for 30 minutes. Bacteria-free glass coverslips were employed as negative controls.

These BIs were placed inside sterile petri dishes in a cell culture hood, transported to our UVGI device, and then exposed to UV-C light in our custom built UV-C light apparatus using Philips G30T8 bulbs (see above). The BI-containing dishes were positioned on a platform positioned directly in the midpoint of the array (where FFRs would be positioned), directly facing the top UV-C array, with petri dish lids removed. Cumulative UV-C doses ranged from 6 to 1000 mJ/cm$^2$, as measured by a calibrated UV-C specific light meter (ILT2400-UVGI; International Light Technologies, Inc., Peabody, MA). After UV-C irradiation, petri dishes were covered, and the BIs were transferred to a biosafety cabinet where they were placed in conical tubes (50 mL) with 5 mL of LB. The loosely-capped tubes were placed in an orbital shaker at 37 C and 250 RPM. After 16 hours of growth, the $OD_{600}$ of each sample was measured (Amersham Biosciences Ultrospec 10 spectrophotometer). *E.coli* amounts were obtained by comparing the $OD_{600}$ for each sample with the $OD_{600}$ from similarly and simultaneously prepared standards derived from a range of non-irradiated control coverslips that had been plated with serially diluted stock *E. coli*. Hydrogen peroxide vapor was used as a decontamination positive control with which to compare UV-C. These BIs were similarly prepared on glass coverslips as above, and then wrapped in sterile Tyvek and placed in a Biosafety Level 3 laboratory at the VA Portland HealthCare System. A negative control was a placed in the room. One positive and one negative control were placed in the hallway and were never exposed to HPV. In addition, hydrogen peroxide chemical indicators (Steris VERIFY HPU Chemical Indicator) were placed throughout the room. HPV was generated using a Bioquell Clarus C and 30% hydrogen peroxide (Fisher). The HPV cycle was 30 minutes of conditioning, pre-gassing, 45 minutes of gassing at 8 gram/minute, 60 minutes of gassing-dwell at 8 gram/minute, followed by an aeration phase. An additional aeration unit was also used. Peak levels of hydrogen peroxide were >400–600 PPM. We used a PortaSens II handheld detector to verify the concentration of hydrogen peroxide was < 1ppm before entering the room. After treatment with HPV, BIs were processed as described above and compared with the $OD_{600}$ from samples that had not been exposed to HPV.

## Results

### UVGI dose measurements

Our UVGI chamber design consisted of two planar arrays of UV-C bulbs facing each other, producing a strong and uniform field of UV illumination, and designed to allow for simultaneous decontamination of multiple masks from both sides. Irradiance diminishes as one moves away from a light source, as described by the inverse square law. This relationship means that shorter inter-array distances provide stronger irradiance (and possibly greater rates of decontamination and throughput), but result in a greater difference in irradiance between the proximal and distal aspects of each FFR facing each array. Our intent was to provide a strong irradiance with minimal non-uniformity between the arrays over the region occupied by the N95s. Between 28–42 cm from each array (the space occupied by the hanging masks directly between the arrays), our arrangement produces UV-C irradiance of 0.47–0.78 mW/cm$^2$ (**Fig 2A–2C**). This range corresponds to the closest and furthest FFR components facing each identical array (proximal and distal regions), as shown in **Fig 2C**. This irradiance provides a dosage of 28–47 mJ/cm$^2$ in one minute, and 1.0–1.6 J/cm$^2$ in approximately 30

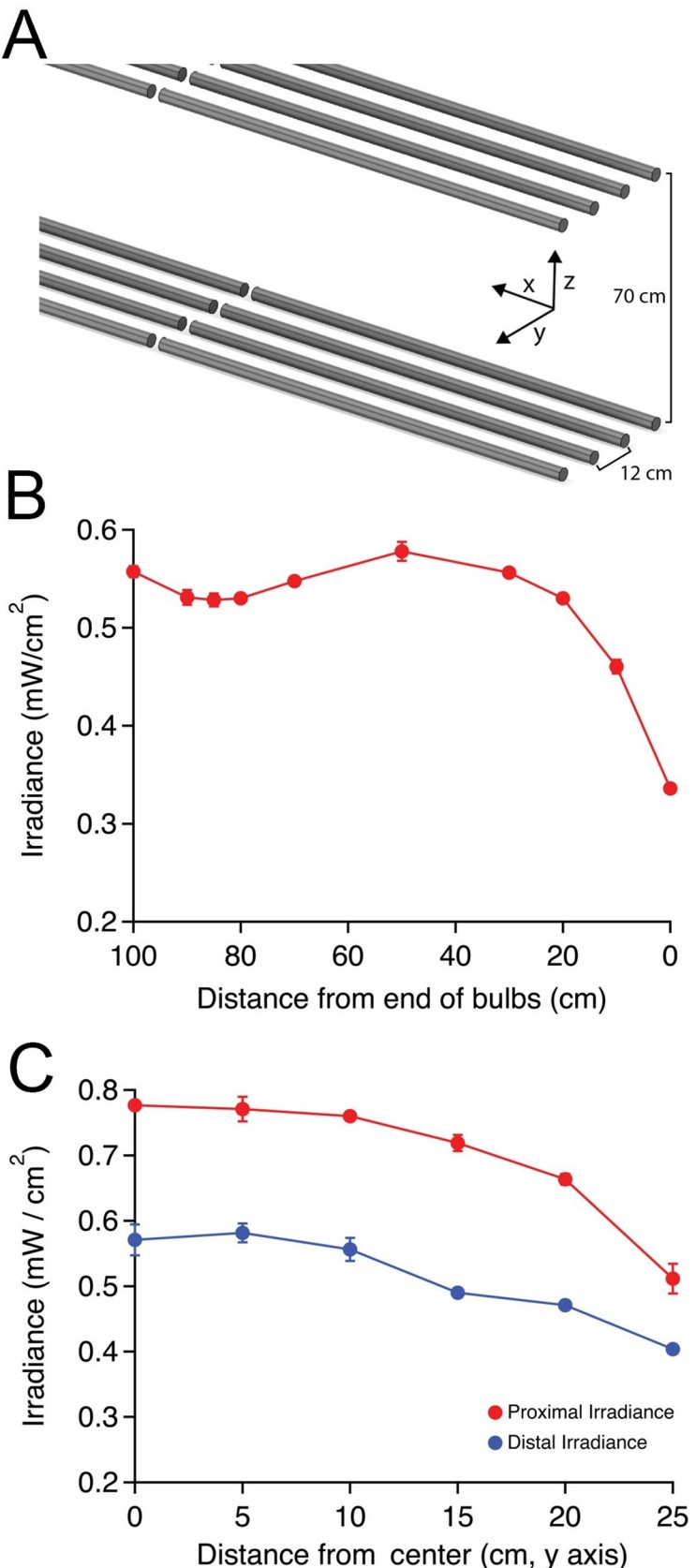

**Fig 2. UV-C light intensity measured within the array.** A. Schematic demonstrating array dimensions and axes. B. Light intensity variation along longitudinal stretch of each array beginning at the terminal tombstone (= 0 cm) moving inwards along the x axis between the center bulbs (mean mW/cm$^2$ ± standard deviation, n = 3 at each location). The slight decrease in irradiance at 85–90 cm corresponds to the region between end-to-end bulbs. C. Light intensity variation moving outwards from the center of the linear array (along the y axis), measured at x = 45 cm. Measurements were recorded with dosimeter facing the array at the proximal and distal vertical extents of each hanging FFR (in this case, z = 28 and 42 cm from array), demonstrating decreased light irradiance at both levels when moving away from the middle of the exposure zone (n = 4 measurements per location ± standard deviation).

minutes. Irradiance was observed to drop off markedly in the outermost 10 cm of the bulb array in the longitudinal direction and beyond 20 cm from midline in the transverse direction (**Fig 2B and 2C**; p < 0.01 for outermost locations compared with any other location; repeated measures ANOVA with post-hoc Tukey's test), and these regions are not used for decontamination. Cycle time was adjusted such that the minimum dosage within the FFR zone would reach 1.0 J/cm$^2$ (which corresponded to irradiance at outermost edges of the region), which our facility chose as a minimum UV-C dose for mask decontamination, and which is consistent with evidence and recently released guidance from governmental and non-governmental agencies [13, 14].

## Germicidal activity verified using locally produced biological indicators

Although we verified the UV-C dosage produced by our array by a specific UV-C detector, additional evidence of functional efficacy of novel decontamination processes is typically obtained through the use of surrogate biological indicators. Biological indicators utilize micro-organisms with a defined resistance to a sterilization process [15]. Various organism classes may be ranked in terms of resistance to decontamination, with bacterial spores ranked amongst the most difficult to kill organisms, while enveloped viruses such as SARS-CoV-2 are ranked amongst the most susceptible to standard disinfection processes [16, 17]. Due to supply disruptions reducing the availability of commercially produced BIs, we developed our own BIs using *E. coli*. *E. coli* was selected because it is much more resistant to decontamination than enveloped single-stranded RNA viruses [17], and provides a faster readout than *G. stearothermophilus* spores, which require days of incubation post-decontamination.

The sensitivity to of *E.coli* to UV decontamination was measured via the optical density of broth after incubation with irradiated, or control non-irradiated, *E.coli*-coated glass discs (Methods; **Fig 3A and 3B**). The *E.coli*-coated glass discs were exposed to discrete doses of UV-C, and subsequently incubated overnight in LB. The optical density of the LB after incubation, which is proportional to bacterial number [18], was plotted against the UV-C irradiance level (**Fig 3C**). These data demonstrate that doses as little as 6 mJoule/cm$^2$ reduced the optical density by ~50%. To characterize the lower limit of detection, we prepared *E. coli*-coated glass discs under identical conditions, but with concentrations of *E. coli* that had been serially 10-fold diluted. We found that even after 5 serial dilutions (1/100,000) the optical density of non-irradiated samples was higher than we observed following an irradiance of 30 mJ/cm$^2$ (Fig 3D). These data indicate that 30 mJoule/cm$^2$ irradiance provided $\geq$ 5-log reduction (99.999%) of viable *E.coli*, which is an organism more resistant to UV-C than single-stranded RNA viruses such as SARS-CoV-2. Our positive controls utilized hydrogen peroxide vapor (HPV) as a gold standard for decontamination, and similarly demonstrated $\geq$ 5-log reduction of *E.coli*.

## Alternative chamber designs

Prior to construction of our UVGI device, we investigated alternative chamber designs, and will briefly describe them here. First, we re-purposed standard cell culture biosafety hoods,

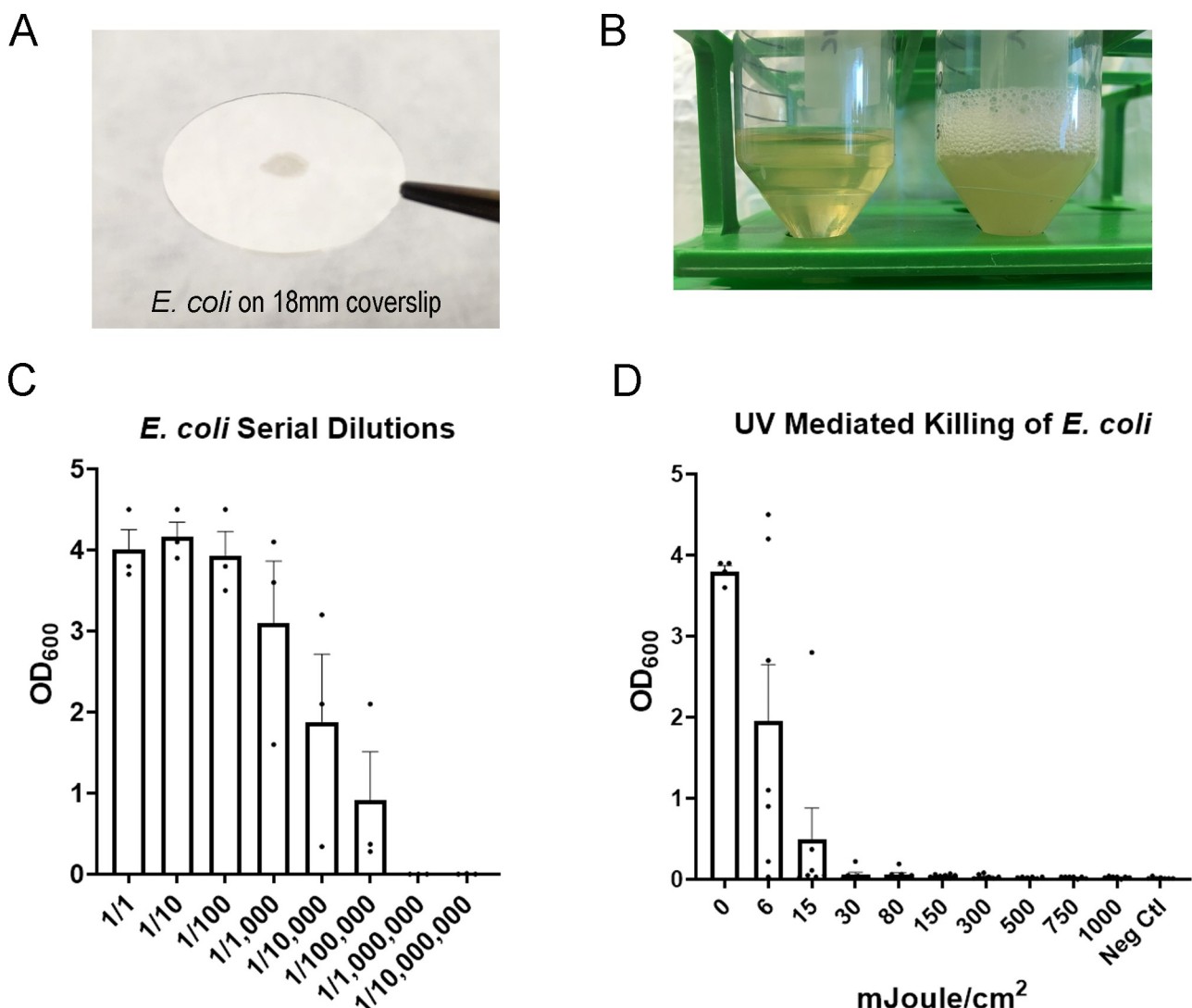

**Fig 3. UV-C light is germicidal to homemade *E. coli* biological indicators.** A. A homemade *Escherichia coli* (*E. coli*) Biological Indicator (BI) shown a on glass coverslip. B. Homemade *E. coli* BIs were incubated for 16 hrs in an orbital shaker. Conical on the left is a negative control (no *E. coli* on disk) and conical on the right is a positive control (*E. coli* on disk). C. Serial dilutions of *E. coli* on glass coverslips. The coverslips were incubated as described above and the Optical Density at 600 nm ($OD_{600}$) was measured. The mean and SEM are shown. Data are pooled from 3 separate experiments. D. Homemade BIs were exposed to discrete doses of UV-C ($mJ/cm^2$) and then incubated as described above. $OD_{600}$ was measured and plotted. The mean and SEM are shown. Data are pooled from separate experiments (0 $mJ/cm^2$: n = 4; 500 $mJ/cm^2$: n = 6; all other doses: n = 7).

which are widely available in facilities with biomedical research labs and which have been previously evaluated for N95 FFR decontamination by others [19]. These unmodified biosafety hoods (SterilGARD, The Baker Company, Sanford, ME) contain a single UV-C bulb in its stock configuration, as well as a UV-C impermeable sliding glass door to protect staff. The bulb in our hood was a GE G64T5 65W bulb, with an expected light intensity of 0.26 $mW/cm^2$ at 1 m distance. However, as irradiance decreases with bulb age, we performed several measurements of irradiance in our hood. We measured an irradiance of 0.18 $mW/cm^2$ of UV-C intensity at the working surface 60 cm directly below the bulb and 0.13 $mW/cm^2$ at the working surface at the front of the hood (75 cm from the bulb). A raised platform 30 cm from the center of the bulb received 0.34 $mW/cm^2$. While readily available, these units did not permit

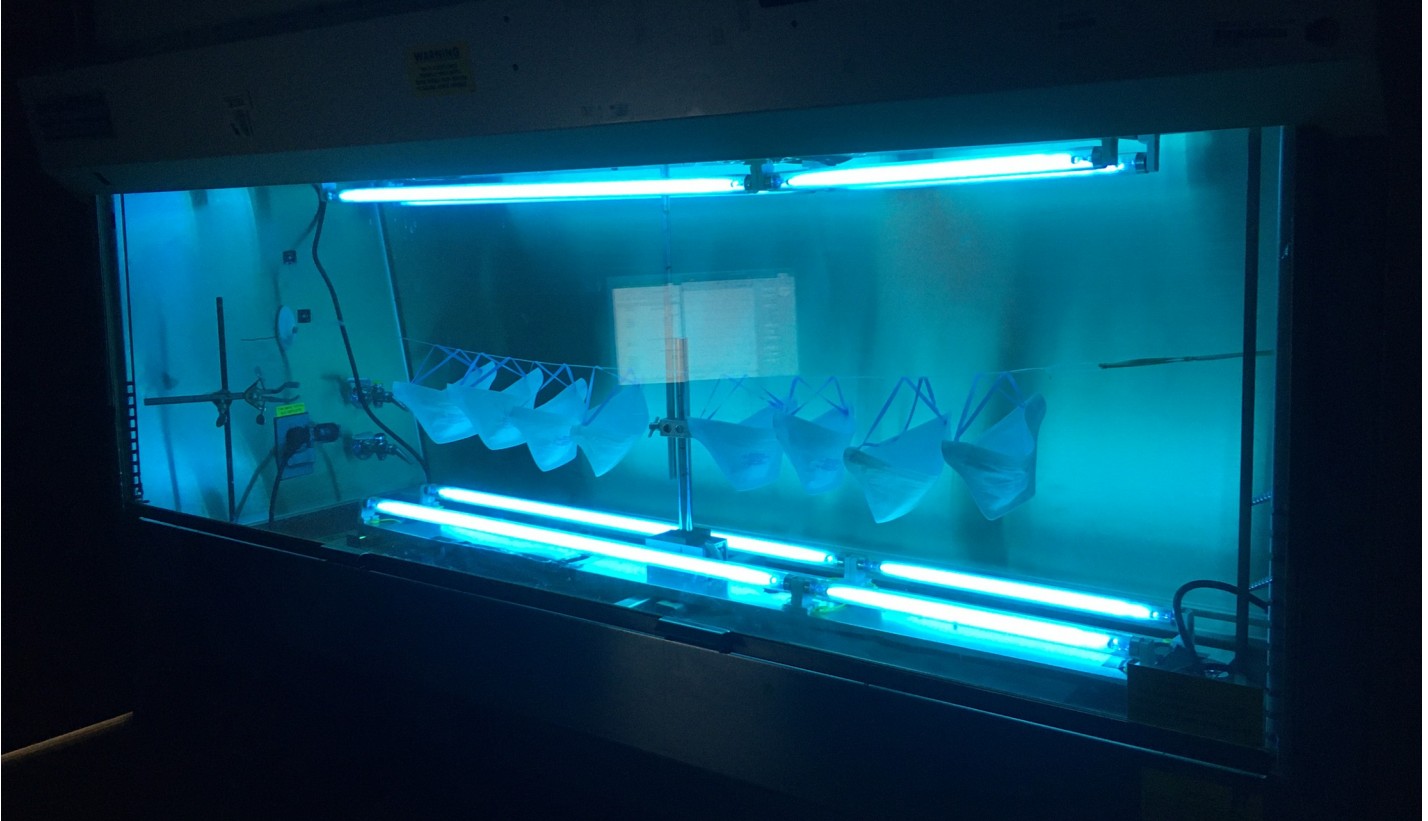

**Fig 4. UVGI decontamination chamber constructed within a modified cell culture biosafety hood.** Two identical four-bulb UV-C arrays were placed facing each other within the hood, and were used to decontaminate masks prior to construction of the larger array (Fig 1).

both surfaces of an N95 FFRs to be irradiated simultaneously, complicating the decontamination process, and the low intensity of the UV-C also necessitated prolonged irradiance (> 4 hours to reach the necessary 1 J/cm$^2$ dose for each surface), which we deemed impractical for high throughput [19].

In a subsequent refinement, additional bulbs (Philips TUV G30T8 30W; see above) were placed in the upper and lower parts of the cabinet, to allow simultaneous UVGI decontamination of both surfaces of each N95 mask, improving the performance of the device (**Fig 4**). The intensity of UV-C at the vertical midpoint of the chamber was 0.66 mW/cm$^2$ in either direction, but was uniform moving laterally until the last 25 cm of each bulb run. Thus, depending on the particular bulbs available at any individual institution, biosafety hoods could be safely modified to accommodate additional light sources to provide bidirectional irradiation. In addition, the use of UV-C reflective material such as household aluminum foil [20] to surround the decontamination chamber could not only reduce the processing time, but also improve the incident angles on FFRs, perhaps increasing efficacy. Although this was not formally evaluated in any of our designs, it has been incorporated in other UVGI systems [21].

## Discussion

The burden on healthcare systems caused by the COVID pandemic has necessitated the adaption of new approaches to ensure availability of PPE for healthcare workers. However, even decontamination strategies typically require resources and supplies that can be difficult to

obtain during global supply line disruption. Our description of simple and cost-effective methods to develop and test a decontamination chamber using locally available components may be of use to other institutions to attenuate the impact of shortages of PPE during the pandemic.

## Limitations and risks of UV-C decontamination

Although UV-C-based N95 FFR decontamination is effective and can be achieved with a variety of either commercially available or custom-built chamber designs [2–6, 10, 22, 23], it does have some limitations. First, these systems can be labor intensive, as they require careful FFR positioning within the UVGI array, such that all surfaces of the FFR are exposed to UV-C irradiation. Second, certain N95 FFR models include the potential requirement for a secondary decontamination method for the straps [5, 24], as this component of each FFR is more variably decontaminated by UVGI. Additionally, although UVGI has been carefully validated for a variety of FFR models, many newer FFR models, including non-surgical N95s or internationally-sourced KN95 FFRs, have not been formally evaluated in terms of their penetration to UV-C light, and thus might be variably decontaminated, as UV-C transmission through N95 layers is material-dependent [25]. Although effective UV-C decontamination doses have been established for many widely available FFR models [2, 5], for FFRs that have not been formally tested, prior work has proposed methodologies for evaluating dose requirements for FFR decontamination [25, 26]. Additionally, although manufacturers and the CDC continue to recommend new FFRs whenever feasible, some have evaluated the effects of various decontamination methods on FFR integrity [27], and could potentially be contacted for reprocessing guidance in crisis situations with low FFR availability. Finally, although prior work specifically evaluated FFR decontamination in regards to enveloped viruses, including SARS-CoV2, more resistant pathogens, such as bacterial spores, might not be inactivated with a 1 J/cm$^2$ dose [13, 24], and thus FFRs intended for decontamination are user-specific, and are often labeled so that they can be returned to the original wearer, to prevent cross-contamination between users. Other decontamination methodologies such as moist heat [7, 28, 29], or a combination of moist heat and UV-C, could be considered as alternative approaches to FFR decontamination [30], and may still retain implementation advantages over vaporized hydrogen peroxide [4], which is very effective but also requires expensive equipment and dedicated facilities that might be difficult to obtain in resource-constrained settings.

Of note, we noticed a nutty/smoky odor when masks were donned shortly after UV-C decontamination. This did not affect fit-testing and was noticed to "off gas" spontaneously after several hours. Although this odor has not been fully characterized, a limited analysis performed by Applied Research Associates demonstrated that levels of 62 volatile organic compounds were either undetectable or several orders of magnitude below permissible exposure limits [24]. We recommend that off-gassing time be considered in UVGI FFR decontamination protocols when feasible. Finally, some UV-C bulbs also produce ozone, which is hazardous and should be appropriately vented.

Finally, some of the limitations of our design relate specifically to its relatively low throughput (as designed, 30 minutes for each batch of ~20 FFRs), and current design requiring in-room changes of FFRs (removal of decontaminated FFRs and hanging of contaminated FFRs). Throughput could be increased with a larger array (to allow more FFRs to hang simultaneously) and reflective material (to increase irradiance and decrease cycle time). We also considered a rolling cantilevered rack that could be loaded and unloaded outside of the chamber, to minimize inter-batch delay, but to facilitate that we would recommend separate doors into the chamber so that decontaminated FFRs could leave via the "clean" exit door, while

contaminated FFRs could enter via a separate door, which would help maintain an appropriate processing workflow and decrease the potential for cross-contamination. Finally, we would recommend that any system be implemented with auto-shutoff features, to turn off the UV-C arrays when staff enter the chamber, to prevent accidental hazardous exposures.

### Benefits of *E. coli* as a BI

To validate decontamination, we explored the use of homemade Bis, which offer several advantages. First, the *E. coli* based BIs described in this report are relatively easy to make and assay with basic laboratory equipment. Second, these BIs serve as a useful surrogate for viral killing because *E. coli* is more difficult to kill than viruses such as SARS-CoV-2 the *E.coli* read-out provides results more rapidly than growing viruses or *G. stearothermophilus*. Third, they are suitable for validating both UV and HPV decontamination methods.

Using homemade BIs, we demonstrated at least 5-log killing efficiency with UV-C doses of 1 Joule/cm$^2$ or an HPV cycle reaching >400 PPM. Since viruses are less resistant to decontamination than *E. coli* [17], this indicates that true viral killing efficiency in our experiments is likely greater than 5-logs. Although a 3-log reduction in coronavirus is considered an acceptable level of decontamination of hard surfaces in healthcare settings [31, 32], N95 FFRs are only rated to block >95% of all infective particles. It is not known whether the infectious risk of FFRs after clinical use derives from the risk of touching a contaminated FFR, in which case it might be treated as a surface, or from potential inhalation of viral particles liberated from FFR fibers during breathing. However, in a crisis situation, an FFR which has been decontaminated and retains its filtering capacity is clearly preferable to either providing clinical care without an FFR or with a non-decontaminated one, so even in the absence of clinical data it remains prudent to pursue decontamination strategies.

One limitation of *E. coli*-based BIs on glass coverslips relates specifically to the alternative medium (glass vs. synthetic N95 fibers) as well as application method, since contaminated FFRs could have been contaminated within deeper layers of the FFR by aerosolized virus-containing particles. Although UVGI-mediated decontamination of FFRs exposed to aerosolized viral particles has been previously validated [2, 5, 7], the ability to test our system using aerosolized virions is beyond our capacity (and beyond the capacity of most hospital systems), and thus our UVGI process design relies on these prior data in combination with validated surrogate measures. Similarly, as each individual FFR cannot be tested without destroying it, the use of these surrogate measures (UVGI irradiance, biological indicators, and prior studies) is critical in implementing these processes. As has been previously emphasized, FFR decontamination is only to be recommended when new FFRs are unavailable, as FFRs begin to degrade in performance after multiple don/doff cycles [33].

To determine the feasibility of making these BIs in resource limited settings, we performed a cost analysis. We excluded fixed costs for a spectrophotometer, shaking incubator and an autoclave. Overall, we found the cost of our process was 1.89 dollars per BI. However, if reusable conical tubes, coverslips and cuvettes are used, the cost decreases to 2.6 cents per BI.

### Conclusions

When supply lines may adversely impact PPE availability, a simple, cost-effective approach can be utilized to construct a chamber using locally-available components for N95 respirator decontamination. These maneuvers can help protect healthcare workers and hence sustain healthcare delivery services during periods when new respirators are unavailable. In addition to the development of a chamber for UV-C decontamination, successful implementation

requires collaboration within the institution to ensure the smooth collection, processing, and storage of decontaminated respirators. These other steps are outlined in the S1 Appendix.

## Supporting information

**S1 Appendix.**
(DOCX)

**S1 Data.**
(XLSX)

## Acknowledgments

We would like to thank David Kagen and Sahana Misra for convening our decontamination workgroup; Darwin Goodspeed for consistent logistic support and encouragement; the VA Portland Electrical Shop and Facilities Management staff for fantastic design implementation and construction assistance; Sherri Atherton (Infection Control) and Ky Dehlinger (Veterinary Medical Unit) for advice; David Cohen and Archie Bouwer for allocation of research space and equipment; Julie Guichot and Oscar Gonzalez for assistance with fit-testing; Sara Frazier, Larry Huebner, Erwin Scully, and Mike Fischer for FFR collection and storage and protocol development; John Dodier for facilities support, Christopher Trapp and the electrical shop for construction assistance, Jim McCarthy for HVAC construction, and Michele Dollar, Grace Chien and Esther Sung for logistic support. We would also like to thank Amy Herr and Alisha Geldert for helpful discussions and comments on this manuscript, and Peter Schnell for assistance with figure preparation. The contents of this manuscript do not represent the views of the U.S. Department of Veterans Affairs or the United States Government.

## Author Contributions

**Conceptualization:** Eric Schnell, Elham Karamooz, Melanie J. Harriff, Jane E. Yates, Stephen M. Smith.

**Data curation:** Eric Schnell, Melanie J. Harriff, Stephen M. Smith.

**Formal analysis:** Eric Schnell, Elham Karamooz, Melanie J. Harriff, Stephen M. Smith.

**Investigation:** Eric Schnell, Elham Karamooz, Melanie J. Harriff, Stephen M. Smith.

**Methodology:** Eric Schnell, Melanie J. Harriff, Jane E. Yates, Stephen M. Smith.

**Project administration:** Eric Schnell, Jane E. Yates, Christopher D. Pfeiffer, Stephen M. Smith.

**Supervision:** Eric Schnell.

**Validation:** Elham Karamooz.

**Writing – original draft:** Eric Schnell, Elham Karamooz, Melanie J. Harriff, Stephen M. Smith.

**Writing – review & editing:** Eric Schnell, Elham Karamooz, Jane E. Yates, Christopher D. Pfeiffer, Stephen M. Smith.

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
