## [Decision Letter · Decision Letter 0]

3 Jun 2021

PONE-D-21-08724

Construction and Validation of an Ultraviolet Germicidal Irradiation System using Locally Available Components

PLOS ONE

Dear Dr. Schnell,

Thank you for submitting your revised manuscript to PLOS ONE, and for your patience as we sought input from external reviewers. After careful consideration, we feel that it has merit but does not fully meet PLOS ONE’s publication criteria as it currently stands. Therefore, we invite you to submit a revised version of the manuscript that addresses the points raised during the review process.

The manuscript was assessed by three external reviewers, whose reports are appended to this letter. As you will see from the reports, the reviewers' comments are broadly positive but they also raise some important concerns which will require clarifications within the methods section and further discussion of some limitations of your study. Please respond to each of the points raised by the reviewers in a response-to-reviewers document, and modify your manuscript accordingly. 

One small point: I see reviewer 1 has suggested moving some content in and out of the appendices, and a title change. In my opinion these changes are optional, so I leave it to you to decide whether you think they improve the overall flow of the manuscript and therefore whether to include them.

We look forward to receiving your revised manuscript.

Kind regards,

Dr Joseph Donlan

Senior Editor

PLOS ONE

Journal Requirements:

[No external funding was explicitly received in support of this work; but the authors would like to acknowledge grant support by the Department of Veterans Affairs, Veterans Health Administration, Office of Research and Development, Biomedical Laboratory Research and Development Merit Review Awards I01-BX004938(ES), I01 CX001562(MH), I01-BX002547 (SMS), a Department of Defense CDMRP Award W81XWH-18-1-0598 (ES), an NIH NINDS 1R21NS102948 (ES), an NIH R01 AI129976 (MH) and an 1R21AI151079-01 (EK). The contents of this manuscript do not represent the views of the U.S. Department of Veterans Affairs or the United States Government.]

 [The author(s) received no specific funding for this work.]

Reviewers' comments:

Reviewer's Responses to Questions

**Comments to the Author**

1. Is the manuscript technically sound, and do the data support the conclusions?

Reviewer #1: Partly

Reviewer #2: Yes

Reviewer #3: Yes

2. Has the statistical analysis been performed appropriately and rigorously? 

Reviewer #1: No

Reviewer #2: Yes

Reviewer #3: Yes

3. Have the authors made all data underlying the findings in their manuscript fully available?

Reviewer #1: Yes

Reviewer #2: Yes

Reviewer #3: Yes

4. Is the manuscript presented in an intelligible fashion and written in standard English?

Reviewer #1: Yes

Reviewer #2: Yes

Reviewer #3: Yes

5. Review Comments to the Author

Reviewer #1: The article makes up for an interesting read in context of its goal of locally developing an in-house UV-C based decontamination system for reprocessing of FFRs. However, it fails to address the most important issues while considering a UV-C based system for reprocessing of FFRs i.e. Decontamination Efficacy & available dose of UV-C to the electret media (filtering layer). Biggest critique of UV-C based FFR reprocessing system is that model specific doses need to be established before it can be recommended for widespread use, an issue which this study doesn’t address at all.

The authors have checked the microbicidal efficacy using Escherichia coli as a surrogate micro-organisms which though not ideal but is acceptable. However, this knowledge about UV-C being germicidal is well known and already established by multiple studies. What is novel about the study is how they have integrated this indigenous system into their workflow which has been provided in the Appendix.

Hence, I would suggest the authors that they should reformat the manuscript by putting the Appendix in the main manuscript and the Germicidal efficacy part can be given as Appendix.

Minor Corrections:

1. “The widely used Spaulding Classification indicates true killing efficiency is greater that 5-logs”: Spaulding classification categorizes items into critical, semicritical and non-critical. It doesn’t tell what should be the log efficiency for killing of micro-organisms. In any case, authors should provide proper reference

2. In Figure 3, authors should provide description of abbreviations.

3. In figure 3, the following is not clear: “Data are pooled from separate experiments (0 mJ/cm2: 2 experiments, n=4; 500 mJ/cm2: 2 experiments, n=6; all other doses: 3 experiments, n=7)” What do they mean by Experiments?.

4. The authors should include a separate paragraph for limitations of the system as well as the study in the Discussion section

5. 5. In title “validation” word should not be used as effect of various variables associated with the delivery of UV-C dose on FFRs is not evaluated. A suitable title can be “Construction & Implementation of an Indigenous UVGI System for Reprocessing of FFRs”.

Reviewer #2: I applaud the authors on a well done manuscript. The references and figures are appropriate and everything is well explained. The constructed UV cabinet is a good low cost alternative for decontaminating FFR in a crisis situation. I do not have any criticisms to offer. Perhaps one thing that can be added is a sentence to check with the FFR manufacturer to see if any additional decontamination guidance is available from them.

Reviewer #3: Dr. Schnell et al. seem to be outlining the relatively simple steps that can be taken to develop and operate an effective UV disinfection system for filtering facepiece respirators during initial stages of a diseases pandemic. This information needs to be understood by more people. While there are multiple ways to develop effective and safe UV disinfection systems for this purpose, information like that presented in the manuscript can help readers understand that it can be done when it seems like everything is going wrong in the world. If we use early in the COVID-19 pandemic as an example, whether we want to admit it or not, healthcare workers were disinfecting respirators for their reuse in multiple ways. Many of the ways employed by individual workers and entire hospitals were likely less effective than the UV disinfection system described here. Personally, I would have designed it differently. In particular, I would have enclosed it with UV-reflective surfaces as much as possible. As it is, there is a lot of wasted UV energy that could have been used to enhance disinfection, lessen disinfection time, or both. An enclosed system also would be safer for those workers responsible for operating the system. Still, the important thing is that the system as described worked. In the throes of a pandemic, things don’t need to be perfect ore even pretty, they just need to do the job. The authors should be commended for their efforts.

The authors have been very responsive to comments from previous reviewers. While I agree with those reviewers, that the work would be strengthened if respirators had been exposed to real coronaviruses (or even good surrogates) as part of the work, the results presented here are still valuable. As one reviewer pointed out, there is a large variability in the UV dose it takes to disinfect different respirator models exposed to the same organism. Similarly, if multiple laboratories did their own experiments to determine what that dose actually is, there is no doubt there would be multiple right answers. Much of the work in the UV field is done with surrogates and/or results from one study are applied to other studies. The authors are not doing anything different in this manuscript. The key is to use a very conservative UV dose so there is no doubt. That is what the authors proposed (and what the FDA has stated).

Ultimately, I think this article is worth publication as is. With that said, there are a few issues where small changes to the manuscript would help:

1.) Were the UV lamps burned-in for at least 100 hours prior to any reported test results? UV lamp output is quite variable in that first 100 hours, so that should have been done.

2.) It is not clear exactly how the coverslip bioindicators were positioned in the UV array during testing. It seems like there were maintained in the petri dishes without the lids, which meant they must have been kept flat. More detail on that step would help.

3.) As part of the discussion, the authors do well to describe limitations of the work. Are there any lessons learned from the work that readers should know about? What changes to the UV disinfection chamber would you make knowing what you know now?

6. PLOS authors have the option to publish the peer review history of their article (what does this mean?). If published, this will include your full peer review and any attached files.

Reviewer #1: **Yes: **Ayush Gupta

Reviewer #2: No

Reviewer #3: No

---

## [Author Response · Author response to Decision Letter 0]

23 Jun 2021

Response to Reviewers

Overall, we are pleased that our reviewers were broadly supportive of our manuscript, referring to our work as commendable, “well done”, and an interesting read, with solid appreciation of the goals and importance of this work in crisis situations with limited resource availability. Of note, two of our three reviewers stated that the paper should be published in its current form, although they did make minor suggestions that we have incorporated into our revision. The other reviewer was also generally supportive of our manuscript, although their primary critique was that we did not address the specific doses of UV-C required for specific Filtering Facepiece Respirator (FFR) models. However, as noted in the text, our manuscript explicitly does not attempt to validate FFR decontamination per se, and we agree (and state) that UV-C doses are model specific and in many cases have already been described in previously published literature, and we have provided some additional information in that regard. We still feel, however, that our description of how to deliver UV-C irradiation using locally available equipment, with the data provided on dose and validation, remain as an important contribution worthy of publication.

We have made additional revisions to clarify this issue, as well as several additional minor corrections as suggested by the reviewers, which we believe have further improved the manuscript. Replies to reviewer comments are in blue text, with changes to the manuscript described in bold. We also would like to amend our Funding Statement as suggested in Dr. Donlan’s letter, and this text is given below in red font.

Major point.

Reviewer #1: The article makes up for an interesting read in context of its goal of locally developing an in-house UV-C based decontamination system for reprocessing of FFRs. However, it fails to address the most important issues while considering a UV-C based system for reprocessing of FFRs i.e. Decontamination Efficacy & available dose of UV-C to the electret media (filtering layer). Biggest critique of UV-C based FFR reprocessing system is that model specific doses need to be established before it can be recommended for widespread use, an issue which this study doesn’t address at all.

Our contribution is to detail the construction and dose validation of a device to deliver UV-C light (to FFRs or other items) using widely available components, and simple instructions on how to potentially implement a decontamination process. We have attempted to avoid implication that our device has been validated for FFRs per se by leaving this out of the title, and have instead focused on demonstrating that the device delivers germicidal irradiation, at quantifiable doses, in a manner that could be used to effectively decontaminate FFRs using doses evaluated in many previous studies. 

We already address the issue with FFR-specific doses with this phrase already in discussion: “Additionally, although UVGI has been carefully validated for a variety of FFR models, many newer FFR models, including non-surgical N95s or internationally-sourced KN95 FFRs, have not been formally evaluated in terms of their penetration to UV-C light, and thus might be variably decontaminated, as UV-C transmission through N95 layers is material-dependent [21].” However, to expand on this issue, we have now provided additional references and suggestions which could potentially assist in helping determine specific dose ranges for newer FFR models in the discussion. Importantly, we also emphasize that FFR decontamination are only intended for crisis situations in which standard supply lines and resource availability have become prohibitive (and life threatening).

Minor points

Reviewer #1: 

The authors have checked the microbicidal efficacy using Escherichia coli as a surrogate micro-organisms which though not ideal but is acceptable. However, this knowledge about UV-C being germicidal is well known and already established by multiple studies. What is novel about the study is how they have integrated this indigenous system into their workflow which has been provided in the Appendix. Hence, I would suggest the authors that they should reformat the manuscript by putting the Appendix in the main manuscript and the Germicidal efficacy part can be given as Appendix.

We appreciate our reviewer’s suggestion on the organization of this manuscript, but in our prior review process, demonstration of germicidal efficacy of our specific system was requested, along with a recommendation that the non-data-supported systems workflow recommendation might be best omitted (or placed into an appendix). Additional testing of our indigenous UVGI decontamination system with surrogate biological indicators adds both an additional level of validation (beyond simple UV-C irradiance/dose measurement), and also provides an additional method through which resource-constrained systems might be able to locally test their system efficacy. Thus, we would prefer to keep the manuscript organization as it currently stands. We agree with the reviewer that the gold standard validation (actual SARS-CoV2 on FFRs) would be ideal, but this assay is rarely feasible outside of BSL-2 laboratory conditions, and thus the use surrogate organisms/measures to evaluate relative efficacy is widely employed and accepted.

Minor Corrections:

1. “The widely used Spaulding Classification indicates true killing efficiency is greater that 5-logs”: Spaulding classification categorizes items into critical, semicritical and non-critical. It doesn’t tell what should be the log efficiency for killing of micro-organisms. In any case, authors should provide proper reference

Thank you for pointing this out. We have provided a more appropriate reference in regards to the relative susceptibility of different microorganisms to decontamination in the Methods. Additionally, the question of “how much decontamination is enough” addressed by Spaulding relates to different medical devices, in large part related to how these devices are used (e.g. critical devices that touch internal/sterile human tissues need to be completely sterile, vs. devices that only touch intact skin are notably less critical). We have also have added additional comments relating to this point in the Discussion.

2. In Figure 3, authors should provide description of abbreviations.

Thank you for catching this omission; abbreviations are now defined in the legend.

3. In figure 3, the following is not clear: “Data are pooled from separate experiments (0 mJ/cm2: 2 experiments, n=4; 500 mJ/cm2: 2 experiments, n=6; all other doses: 3 experiments, n=7)” What do they mean by Experiments?. 

The sample size (n) refers to the number of coverslips exposed to each dose of UV-C, and each coverslip was subsequently cultured in its own tube of LB. “Experiments” refers to different batches of coverslips processed in parallel (we ran three separate batches on different dates to increase sample size). Two of the UV-C dose groups were only run twice. The legend was revised to just cite “n” to avoid confusion.

4. The authors should include a separate paragraph for limitations of the system as well as the study in the Discussion section

The limitations inherent to UVGI and reprocessed (vs. new) single-use FFRs is extensively reviewed in the discussion and the references contained therein; but we have expanded our discussion section with an additional paragraph relating to limitations of our specific system and suggestions for improvement.

5. 5. In title “validation” word should not be used as effect of various variables associated with the delivery of UV-C dose on FFRs is not evaluated. A suitable title can be “Construction & Implementation of an Indigenous UVGI System for Reprocessing of FFRs”.

We appreciate our reviewer’s constructive advice on our title, but we actually removed reference to N95s/FFRs and added the word ‘validation’ to the title based on feedback from the prior review of this manuscript at PLOS ONE (which originally had a title quite close to the reviewer’s suggestion). As we believe that it would be inappropriate to imply that our system was specifically validated for FFRs, we would respectfully prefer to keep the title as it stands.

Reviewer #2: I applaud the authors on a well done manuscript. The references and figures are appropriate and everything is well explained. The constructed UV cabinet is a good low cost alternative for decontaminating FFR in a crisis situation. I do not have any criticisms to offer. Perhaps one thing that can be added is a sentence to check with the FFR manufacturer to see if any additional decontamination guidance is available from them.

We appreciate the reviewer’s enthusiasm. Regarding manufacturer guidance, most FFRs are intentionally designed for single-use, and thus most manufacturers have not evaluated or recommended specific decontamination methodologies. We have, however, included this suggestion in the 2nd paragraph of the discussion, as some manufacturers (e.g. 3M) have actually evaluated various decontamination methods (please see reference 27, also added). We also note that although UV-C decontamination is acceptable in maintaining FFR function, 3M continues to not recommend decontamination unless in a crisis situation. 

Reviewer #3: Dr. Schnell et al. seem to be outlining the relatively simple steps that can be taken to develop and operate an effective UV disinfection system for filtering facepiece respirators during initial stages of a diseases pandemic. This information needs to be understood by more people. While there are multiple ways to develop effective and safe UV disinfection systems for this purpose, information like that presented in the manuscript can help readers understand that it can be done when it seems like everything is going wrong in the world. If we use early in the COVID-19 pandemic as an example, whether we want to admit it or not, healthcare workers were disinfecting respirators for their reuse in multiple ways. Many of the ways employed by individual workers and entire hospitals were likely less effective than the UV disinfection system described here. Personally, I would have designed it differently. In particular, I would have enclosed it with UV-reflective surfaces as much as possible. As it is, there is a lot of wasted UV energy that could have been used to enhance disinfection, lessen disinfection time, or both. An enclosed system also would be safer for those workers responsible for operating the system. Still, the important thing is that the system as described worked. In the throes of a pandemic, things don’t need to be perfect ore even pretty, they just need to do the job. The authors should be commended for their efforts.

Thank you for the positive feedback. Although we designed our light array for simplicity and minimal resource utilization, we agree that UV-reflective surfaces would likely not only reduce the processing time, but also improve the incident angles on FFRs, perhaps increasing efficacy. Household aluminum foil is actually quite reflective (~75% for UV-C), often widely available, and has been incorporated in other designs to date, and we have added this suggestion (and references) to our ‘Alternative chamber designs’ section.

The authors have been very responsive to comments from previous reviewers. While I agree with those reviewers, that the work would be strengthened if respirators had been exposed to real coronaviruses (or even good surrogates) as part of the work, the results presented here are still valuable. As one reviewer pointed out, there is a large variability in the UV dose it takes to disinfect different respirator models exposed to the same organism. Similarly, if multiple laboratories did their own experiments to determine what that dose actually is, there is no doubt there would be multiple right answers. Much of the work in the UV field is done with surrogates and/or results from one study are applied to other studies. The authors are not doing anything different in this manuscript. The key is to use a very conservative UV dose so there is no doubt. That is what the authors proposed (and what the FDA has stated).

Ultimately, I think this article is worth publication as is. With that said, there are a few issues where small changes to the manuscript would help:

1.) Were the UV lamps burned-in for at least 100 hours prior to any reported test results? UV lamp output is quite variable in that first 100 hours, so that should have been done.

The UV bulbs were not burned in. We agree that UV-C bulb output can vary between bulbs, within the same bulbs as they age/burn-in, and even during use (in our hands, irradiance increased substantially over the first minute of being turned on). For these reasons, we (and others) have strongly recommended that a UV-C irradiance meter is used to measure total energy delivered for every UVGI run as a quality control parameter. We added a comment regarding the variable output of UV-C bulbs during burn-in as an additional justification for this recommendation.

2.) It is not clear exactly how the coverslip bioindicators were positioned in the UV array during testing. It seems like there were maintained in the petri dishes without the lids, which meant they must have been kept flat. More detail on that step would help.

The coverslip bioindicators were positioned on a platform positioned directly in the midpoint of the array (where FFRs would be positioned), directly facing the top UV-C array, with petri dish lids removed. These details were clarified to the Methods.

3.) As part of the discussion, the authors do well to describe limitations of the work. Are there any lessons learned from the work that readers should know about? What changes to the UV disinfection chamber would you make knowing what you know now?

Thank you. During evaluation and implementation of several different FFR decontamination processes in parallel, we eventually settled on vaporized H2O2 as our primary methodology due to is higher throughput. However, it was also clear that this method is not only very expensive (~$100K set up, at a minimum), time consuming (weeks to construct an H2O2-safe room/facility) requires incredibly high level training/safety protocols/infrastructure, and requires devices that would be almost impossible to source during a crisis. The resource-intensiveness of vaporized H2O2 is already mentioned in the introduction. 

However, during our ongoing work with the UVGI-system, we did learn several lessons relating to throughput and design improvements (including the suggestion relating to the addition of low-cost reflective material). Although several of these insights were originally in the appendix, we have now included some of the lessons learned in a new paragraph in the discussion.

Housekeeping:

Our statement regarding funding information was removed from our Acknowledgements section, but even though this funding was not explicitly intended for this work, we feel that it would be appropriate to mention these sources in our “Funding Statement”. Thank you for this opportunity. If possible, please change the Funding Statement on our online submission form to include the following text:

The authors would like to acknowledge grant support by the Department of Veterans Affairs, Veterans Health Administration, Office of Research and Development, Biomedical Laboratory Research and Development Merit Review Awards I01-BX004938(ES), I01 CX001562 (MH), I01-BX002547 (SMS), a Department of Defense CDMRP Award W81XWH-18-1-0598 (ES), an NIH NINDS 1R21NS102948 (ES), an NIH R01 AI129976 (MH) and an 1R21AI151079-01 (EK).

---

## [Decision Letter · Decision Letter 1]

12 Jul 2021

Construction and Validation of an Ultraviolet Germicidal Irradiation System using Locally Available Components

PONE-D-21-08724R1

Dear Dr. Schnell,

We’re pleased to inform you that your manuscript has been judged scientifically suitable for publication and will be formally accepted for publication once it meets all outstanding technical requirements.

Kind regards,

Jianguo Wang, PhD

Academic Editor

PLOS ONE

Additional Editor Comments (optional):

Reviewers' comments:

Reviewer's Responses to Questions

**Comments to the Author**

1. If the authors have adequately addressed your comments raised in a previous round of review and you feel that this manuscript is now acceptable for publication, you may indicate that here to bypass the “Comments to the Author” section, enter your conflict of interest statement in the “Confidential to Editor” section, and submit your "Accept" recommendation.

Reviewer #1: All comments have been addressed

Reviewer #2: All comments have been addressed

2. Is the manuscript technically sound, and do the data support the conclusions?

Reviewer #1: Yes

Reviewer #2: Yes

3. Has the statistical analysis been performed appropriately and rigorously? 

Reviewer #1: Yes

Reviewer #2: Yes

4. Have the authors made all data underlying the findings in their manuscript fully available?

Reviewer #1: (No Response)

Reviewer #2: Yes

5. Is the manuscript presented in an intelligible fashion and written in standard English?

Reviewer #1: Yes

Reviewer #2: Yes

6. Review Comments to the Author

Reviewer #1: (No Response)

Reviewer #2: I accept the revision based on the author's replies to the original review. This is a valuable paper for communities with low budgets to perform decontamination.

7. PLOS authors have the option to publish the peer review history of their article (what does this mean?). If published, this will include your full peer review and any attached files.

Reviewer #1: **Yes: **Ayush Gupta

Reviewer #2: No

---

## [Editor Report · Acceptance letter]

16 Jul 2021

PONE-D-21-08724R1 

Construction and Validation of an Ultraviolet Germicidal Irradiation System using Locally Available Components 

Dear Dr. Schnell:

I'm pleased to inform you that your manuscript has been deemed suitable for publication in PLOS ONE. Congratulations! Your manuscript is now with our production department. 

Kind regards, 

on behalf of

Dr. Jianguo Wang 

Academic Editor

PLOS ONE